# Performance of Sn-3.0Ag-0.5Cu Composite Solder with Kaolin Geopolymer Ceramic Reinforcement on Microstructure and Mechanical Properties under Isothermal Ageing

**DOI:** 10.3390/ma14040776

**Published:** 2021-02-07

**Authors:** Nur Syahirah Mohamad Zaimi, Mohd Arif Anuar Mohd Salleh, Andrei Victor Sandu, Mohd Mustafa Al Bakri Abdullah, Norainiza Saud, Shayfull Zamree Abd Rahim, Petrica Vizureanu, Rita Mohd Said, Mohd Izrul Izwan Ramli

**Affiliations:** 1Center of Excellence Geopolymer & Green Technology (CeGeoGTech), University Malaysia Perlis (UniMAP), Taman Muhibbah, Perlis 02600, Malaysia; syahirahzaimi25@gmail.com (N.S.M.Z.); mustafa_albakri@unimap.edu.my (M.M.A.B.A.); norainiza@unimap.edu.my (N.S.); shayfull@unimap.edu.my (S.Z.A.R.); rita@unimap.edu.my (R.M.S.); Izrulizwan@unimap.edu.my (M.I.I.R.); 2Faculty of Chemical Engineering Technology, University Malaysia Perlis (UniMAP), Taman Muhibbah, Perlis 02600, Malaysia; 3Faculty of Materials Science and Engineering, Gheorghe Asachi Technical University of Iasi, 71 D. Mangeron Blv., 700050 Iasi, Romania; 4Romanian Inventors Forum, Str. Sf. P. Movila 3, 700089 Iasi, Romania; 5Faculty of Mechanical Engineering Technology, Pauh Putra Campus, University Malaysia Perlis (UniMAP), Perlis 02600, Malaysia

**Keywords:** composite solder, intermetallics, microstructure, ageing, activation energy, lead-free solder geopolymer, geopolymer ceramic

## Abstract

This paper elucidates the effect of isothermal ageing at temperature of 85 °C, 125 °C and 150 °C for 100, 500 and 1000 h on Sn-3.0Ag-0.5Cu (SAC305) lead-free solder with the addition of 1 wt% kaolin geopolymer ceramic (KGC) reinforcement particles. SAC305-KGC composite solders were fabricated through powder metallurgy using a hybrid microwave sintering method and reflowed on copper substrate printed circuit board with an organic solderability preservative surface finish. The results revealed that, the addition of KGC was beneficial in improving the total thickness of interfacial intermetallic compound (IMC) layer. At higher isothermal ageing of 150 °C and 1000 h, the IMC layer in SAC305-KGC composite solder was towards a planar-type morphology. Moreover, the growth of total interfacial IMC layer and Cu_3_Sn layer during isothermal ageing was found to be controlled by bulk diffusion and grain-boundary process, respectively. The activation energy possessed by SAC305-KGC composite solder for total interfacial IMC layer and Cu_3_Sn IMC was 74 kJ/mol and 104 kJ/mol, respectively. Based on a lap shear test, the shear strength of SAC305-KGC composite solder exhibited higher shear strength than non-reinforced SAC305 solder. Meanwhile, the solder joints failure mode after shear testing was a combination of brittle and ductile modes at higher ageing temperature and time for SAC305-KGC composite solder.

## 1. Introduction

Solder alloy is used as an interconnecting material in microelectronic packaging, joining the components to substrate [1]. Development of lead-free solder alloy was ineluctable due to the prohibition of lead usage in the electronic industry implemented by environmental laws. The toxicity of lead has devoted the industry to search for a new generation of solder alloy which is free from lead content. Numerous studies have been carried out in finding a suitable lead-free solder alloy [2,3,4]. Tin-silver-copper (SAC) solder alloy is identified to be one of promising candidate which could replace the conventional lead solder alloy [4,5,6,7,8,9]. Other than being widely available in the market industry, SAC solder alloy also has good mechanical properties, and better solderability with a suitable melting temperature [8,10]. Even so, the growth of interfacial intermetallic compound (IMC) phases, such as Cu_6_Sn_5_ and Cu_3_Sn, at the solder/substrate in SAC solder joints is faster as compared to eutectic tin-lead (SnPb) solder joints, resulting in a thicker interfacial IMC layer [11]. A thicker interfacial IMC layer was responsive to the stress and may worsen the reliability of the solder joints as it could promote cracks to be initiated and propagated. Meanwhile, too thin of an interfacial IMC layer may not form a proper solder interconnection. Therefore, controlling the thickness of the interfacial IMC layer is crucial to improve the reliability of the solder joints.

Recently, composite solder approaches have been recognized as one of the potential methods to improve the reliability of the solder joints since it can provide a marked improvement to the microstructure and mechanical strength. Ceramic materials, such as titanium oxide (TiO_2_), aluminium oxide (Al_2_O_3_), silicon carbide (SiC), cerium oxide (CeO_2_) and silicon nitride (Si_3_N_4_), have been chosen as the ones that could be added into solder alloy, forming composite solder. These ceramic materials work by being dispersed in the matrix of Sn solder alloy and distributed homogenously along the grain boundaries of Sn solder alloys. Extensive studies have been reported on how the ceramic materials could improve the performance of lead-free solder alloys by refining the microstructure, inhibiting the growth of interfacial IMC layer and strengthening the solder joints [12,13,14,15]. Liu et al. [13] hypothesized that, additions of TiO_2_ could inhibit the growth of IMC layer while refining the Cu_6_Sn_5_ grains in Sn-based solder alloys. Moreover, Wu et al. [13] reported the ability of the Al_2_O_3_ in reducing the growth rate of interfacial Cu_6_Sn_5_ in Sn-0.3 Ag-0.7Cu. Meanwhile, Li et al. [16] made an attempt by adding CeO_2_ in SAC solder alloys which resulted in lowering the rate of Cu atoms’ diffusion at the copper/solder interface through a diffusion-controlled kinetic model.

In our previous works [17], kaolin geopolymer ceramic-reinforced Sn-3.0Ag-0.5Cu (SAC305) composite solders were fabricated with various weight percentages (0–2.0 wt%) and investigated in terms of microstructure, mechanical strength and thermal properties. The results proved that, 1 wt% kaolin geopolymer ceramic was able to yield an optimum result in terms of thickness of the IMC layer, refining the microstructure and shear strength. However, the above-mentioned only focused on the effects of kaolin geopolymer ceramics in SAC305 solder alloys in as-reflowed conditions. As the current development of electronic packaging is continuously downsizing the size of the solder joints, the reliability of the joining becomes very important, especially under actual working conditions. During actual working conditions, solder joints are frequently exposed to solid state ageing conditions, such as continuous use of electronic equipment and switching on-off cycles of the electronic devices. These events could facilitate the growth of interfacial IMC layers as the IMCs are sensitive towards the temperature changes and IMC formations could significantly change during this condition. The reactions during solid state ageing differs significantly under influence of several temperature ranges [18]. For example, Cu_3_Sn starts to grow at temperatures below 60 °C and increases at longer duration. Therefore, isothermally age-testing is typically used to simulate the changes of microstructure and performance of the solder joints during the actual working conditions [11,19,20,21].

Therefore, this paper aims to elucidate the effect of the addition of 1 wt% kaolin geopolymer ceramics on the growth kinetics of Cu_6_Sn_5_ and Cu_3_Sn IMCs in SAC305 solder alloy and examine the suppression effect on the IMC layer and mechanical performance subjected to several ageing times and temperatures.

## 2. Materials and Methods

Sn-3.0Ag-0.5Cu (SAC305) solder powders with average particle sizes of 25–45 µm with spherical morphology were used as the solder matrix material. The reinforcement, which is kaolin geopolymer ceramic (KGC) powder, having average particle size of ~18 µm, were used in this research. The fabrication of KGC was preceded with the formation of kaolin geopolymer. Formation of kaolin geopolymer took place as the kaolin was activated with alkaline activator solution and underwent curing process at 80 °C for a day. Then, the kaolin geopolymer was pulverized by using a mechanical crusher and uniaxially compressed using a load of 4.5 tons. The sintering process of a compacted pellet was took place at 1200 °C, forming kaolin geopolymer ceramic (KGC). To obtain KGC with an average particle size of ~18 µm, sintered KGC underwent a ball milling process for about 10 h at a speed of 450 rpm. The ball to powder ratio (BPR) used during the ball milling process was 10:1 in a planetary ball milling machine. 

To fabricate SAC305-KGC composite solders, powder metallurgy with a hybrid microwave sintering method was used. SAC305 with 1 wt% KGC was weighed and homogeneously mixed in an airtight container by using a planetary mill with a speed of 200 rpm for an hour. Then, the mixture was uniaxially compacted with a load of 4.5 tons in a stainless steel mould. The sintering process of compressed pellets were using a hybrid microwave sintering technique at 185 °C under ambient conditions, with an output power of 800 W, 50 Hz microwave oven for 3 min. For comparison, pure SAC305 solder was compacted and sintered by using the same method. To obtain a thin solder sheet with approximately 50 µm thickness, the sintered pellet was cold rolled using a rolling machine. Fabrication of solder balls with a size of 900 µm took place as a 3.0 mm puncher was used to punch a thin solder sheet. The punched solder sheets were then immersed in rosin mildly-activated flux (RMA) and heated on a Pyrex plate at 250 °C using a reflow oven. Then, the sieving of the solder ball with the use of 1 mm and 0.9 mm sieve was done in order to have a constant size of the solder ball.

A slight amount of RMA flux was applied to the solder ball and was placed on a Cu substrate printed circuit board with an organic solderability perspective surface finish with a ball pitch size of 900 µm. Then, the solder ball was reflowed in a F4N desktop reflow oven at 250 °C with time above liquids of 25 s. To study the growth of interfacial IMC, the samples were isothermally aged at temperatures of 85 °C, 125 °C and 150 °C for 100, 500 and 1000 h, which follows the JEDEC standard temperature (JESD22-A103C) [22] The microstructure analysis was conducted by using optical microscope (OM). The elements in the samples were analysed by using energy dispersive spectroscopy (EDS) on cross-sectioned aged samples. The samples were cold mounted with epoxy resin, mechanically grounded on silicon carbide papers and polished with an alumina suspension of 1.0 and 0.3 µm. To discover the details of the microstructure of the solder joint, oxide polishing suspensions (OPS) were used for final polishing. The thickness of the interfacial IMC was calculated based on optical microscope (OM) images taken by using ImageJ software (developed at National Institutes of Health and Laboratory for Optical and Computational Instrumentation, University of Winconsin, Madison, WI, USA). The thickness of interfacial IMC was measured by divided the total area of IMC with a total length of IMC.

Additionally, the strength of the solder joints after isothermal ageing was determined through single-lap shear solder joint test. The single-lap shear solder joint was performed using Instron machine with reference to ASTM D1002 standard [17]. The average shear strength was calculated from a total of five samples for each compositions and ageing conditions. Then, the fracture mode after single-lap shear test was investigated using a scanning electron microscope (SEM) with an energy dispersive X-ray under secondary imaging mode and an accelerating voltage of 20 kV.

## 3. Results and Discussions

### 3.1. Evolutions of IMC of Microstructure

Figure 1 illustrate the microstructure of non-reinforced SAC305 and SAC305-KGC composite solder at the bulk solder prior to reflow soldering. Based on the figure, the microstructure of the solder involves primary IMCs, eutectic and β-Sn area prior to reflow soldering process. Meanwhile, Figure 2, Figure 3 and Figure 4 present the microstructure formations of non-reinforced SAC305 and SA305-KGC composite solders at the solder bulk subjected to isothermal ageing at 85 °C, 125 °C and 150 °C at several isothermal ageing times. Based on the figure, it can be clearly observed that the increase in ageing temperature and ageing time causes the primary IMCs to become coarser and larger in non-reinforced SAC305 solder. On top of this, the coarsening of IMCs in the eutectic area was also observed in non-reinforced SAC305 especially at higher temperature of 150 °C and longer time (as in Figure 5c). However, for SAC305-KGC composite solder sample, the coarsening of IMCs during solid state isothermal ageing was not obvious. This alludes that addition of KGC particles could controls and suppressed the coarsening of IMCs in eutectic area and formation of primary IMCs even during isothermal ageing. The same phenomenon can be found in a study done by Sobhy et al. [23]. As studied by Gain et al. [24], the incorporation of ceramic particles in the solder may alter the diffusivity and chemical affinity which lead to refining of IMCs particles even during solid state isothermal ageing.

The intermetallic compound (IMC) at the interfacial was investigated based on the average thickness and morphology of IMC layer. Figure 6 shows the cross-sectional images of non-reinforced SAC305 and SAC305-KGC composite solders prior to reflow soldering process. Based on Figure 6a, the morphology of interfacial IMC layer in non-reinforced SAC305 solder had an elongated and rounded scallop of Cu_6_Sn_5_. Meanwhile, the morphology of the interfacial IMC layer in SAC-KGC composite solder had rounded scallop Cu_6_Sn_5_ and the formation of elongated scallop Cu_6_Sn_5_ was not observed. According to our previous study, the elongated scallop was formed due to the increase in concentration of Cu atoms from the substrates, diffusing and reacting with Sn, forming elongated scallop Cu_6_Sn_5_ [17]. However, in this research Cu_3_Sn layer was not be observed on the sample of non-reinforced SAC305 and SAC305-KGC composite solders after reflow soldering due to the significantly thinner layer. The Cu_3_Sn phase is known to be formed during the reaction between Sn-Cu/Cu after reflow soldering as reported by Feng et al. [25].

Meanwhile, Figure 7, Figure 8 and Figure 9 show the interfacial IMC layer of non-reinforced SAC305 and SAC305-KGC composite solder subjected to isothermal ageing at temperatures of 85 °C, 125 °C and 150 °C for 100 h, 500 h and 1000 h. Based on the figures, the interfacial IMC layer consists of duplex IMC structures. According to EDX analysis as in Figure 10, the duplex structure consists of light layer (Point 1) which corresponds to Cu_6_Sn_5_ IMC layer. Then, a dark layer (Point 2) was corresponding to Cu_3_Sn IMC layer. In addition, Figure 7 shows the interfacial IMC layer of non-reinforced SAC305 and SAC305-KGC composite solders subjected to a low isothermal ageing temperature of 85 °C for 100 h, 500 h and 1000 h. The samples showed combination of elongated and rounded scallop Cu_6_Sn_5_ with a very thin Cu_3_Sn IMC layer. However, as the isothermal ageing process continues with higher temperatures of 125 °C and 150 °C, the morphology of interfacial IMC layer in non-reinforced SAC305 was formed with some elongated and small rounded scallop as depicted in Figure 8a–c and Figure 9a–c. Meanwhile, in SAC305-KGC composite solder, the interfacial IMC layer grew towards more planar-type morphology with increasing ageing time and temperature as shown in Figure 9d–f. 

The changes in the morphology of Cu_6_Sn_5_ IMC in SAC305-KGC composite solder towards more planar-type was due to the distance at scallop valley was closer to the substrates as compared to distance at the scallop of Cu_6_Sn_5_ peak to the substrates. This occurrence causes a faster Cu diffusion at the scallop valley and, thus, planarizing the morphology of interfacial Cu_6_Sn_5_ IMC [2,11,22,26]. Furthermore, the changes in the morphology also relates with the differences in the Gibs Free energy. The changes in the IMC structures occurred during the isothermal ageing was aims to lower the surface energy. The surface energy associated with scallop Cu_6_Sn_5_ IMC was higher compared to the layered type. The heat produced from the isothermal ageing process causes the surface tension of the Cu_6_Sn_5_ IMC to be unstable. In order to stabilize the surface tension, the excess energy was removed through atoms diffusing, resulting in higher Cu atom diffusion and, thus, producing a layered type of Cu_6_Sn_5_ IMC [27].

To precisely elucidate the growth of interfacial IMC layer on non-reinforced SAC305 and SAC305-KGC composite solders during isothermal ageing, the average thickness of interfacial IMC layer was calculated by using ImageJ software Figure 11 illustrates the bar graph showing the total (Cu_6_Sn_5_ + Cu_3_Sn) average of interfacial IMC thickness subjected to different isothermal ageing temperatures and time. Based on the figure, it can be seen clearly that the total interfacial IMC thickness increased in non-reinforced SAC305 and SAC305-KGC composite solders as the temperature and time increased. However, the total interfacial IMC thickness of non-reinforced SAC305 was higher compared to SAC305-KGC composite solder. Figure 11c illustrates, that the total interfacial IMC thickness of non-reinforced SAC305 at isothermal ageing of 150 °C/1000 h was ~17 µm. Meanwhile, the total interfacial IMC thickness for SAC305-KGC composite solder under the same condition was ~14 µm. Therefore, this result suggests that the incorporation of KGC particles in SAC3055 solder is beneficial in suppressing the interfacial IMC thickness during solid state ageing with the suppression of approximately 15%.

As reported by Wang et al. [28], Cu_3_Sn layer may be formed between Cu_6_Sn_5_ and Cu substrates during reflow soldering process or after ageing. The formation of Cu_3_Sn layer resulted from solid state diffusion. As reported by Mohd Salleh et al. [29], Cu_6_Sn_5_ were rapidly formed during the early reaction between solder alloy and Cu substrate. However, with continuous diffusion of Cu, a layer of Cu_3_Sn will be formed in between Cu_6_Sn_5_ and Cu substrates. As the Cu atoms reach at the interface between the Cu_6_Sn_5_/Cu_3_Sn, the following reaction will occur [2]:

Cu_6_Sn_5_ + 9Cu → 5Cu_3_Sn
(1)

Based on the Equation (1), the reaction of Cu with Cu_6_Sn_5_ cause Cu_6_Sn_5_ to be converted to Cu_3_Sn. Therefore, the Cu atoms diffusion at Cu_6_Sn_5_/solder will be reduced. Thus, the growth of Cu_6_Sn_5_ layer will be interrupted, while the growth of Cu_3_Sn layer will be increased with prolong ageing time and temperature. In this research, Cu_3_Sn layer was only observed after ageing the sample at 85 °C for 100 h. Figure 12 illustrates the average Cu_3_Sn IMC thickness subjected to isothermal ageing at temperature of 85 °C, 125 °C and 150 °C for 100, 500 and 1000 h. From the bar graph, it is clearly shown that, with prolonged ageing time and higher temperature, the thickness of interfacial Cu_3_Sn was increased. The average thickness of Cu_3_Sn IMC in non-reinforced SAC305 solders during isothermal ageing at 150 °C for 1000 h was 2.20 µm, whereas for SAC305-KGC composite solder, the average thickness of Cu_3_Sn IMC was 1.60 µm. The difference about ~24% in the average Cu_3_Sn IMC thickness between non-reinforced SAC305 and SAC305-KGC composite solders under isothermal ageing of 150 °C/1000 h prove that the addition of KGC particles were able to suppress the growth of Cu_3_Sn IMC layer especially at higher ageing temperature and longer time.

Overall, the thickness of total interfacial IMC layer (Cu_6_Sn_5_ and Cu_3_Sn) and Cu_3_Sn in SAC305-KGC composite solder was greatly suppressed under the isothermal ageing process. The suppression in the thickness of interfacial IMC layer was due to the ability of KGC particles across the matrix of solders to hinder the diffusion of Cu at the interfacial IMC layer. The added KGC particles may be absorbed on the surface of IMCs and retarding the further growth of IMC layer. Moreover, as reported by Mohamad Zaimi [17], segregation of KGC particles existed across the molten solder matrix and interfacial layer which supports the ability of KGC particles to alter the growth of interfacial IMC layer.

### 3.2. Growth Kinetics of IMC Layer

Growth kinetics of IMC layer during the solid-state isothermal ageing can be described according to the empirical power-law relationship as following [30,31]: (2)$$\mathrm{Xt}=\mathrm{Xo}+{\mathrm{Dt}}^{n},$$
where Xt is the IMC thickness (m) at ageing time t (s), Xo is the initial thickness of IMC layer after reflow soldering and D is the diffusion coefficient (m^2^/s) and n is the time exponent. The value of time exponent *n* could be considered as an indicator to dictate the controlling mechanism for the growth of IMC layer. If the value of n is closer to n = 0.33, the controlling mechanism for the growth of IMC layer can be described as grain-boundary diffusion. Meanwhile, if the value of n is closer to n = 0.5 and n = 1.0, the controlling mechanism for the growth of IMC layer is described as bulk diffusion-controlled process or an interface reaction rate-controlled process, respectively. 

In this research, the value of the time exponent n for total IMC layer (Cu_6_Sn_5_ and Cu_3_Sn) and Cu_3_Sn can be attained by using linear regression analysis. The values of time exponent *n* for total IMC layer and Cu_3_Sn are presented in Figure 13. Based on Figure 13a,b, the values of time exponent n for total IMC layer in non-reinforced SAC305 and SAC305-KGC composite solders were in the range of 0.45–0.52. The values obtained tend to be closer to 0.5. Thus, it can be considered that the growth of total IMC layer in non-reinforced SAC305 and SAC305-KGC composite solders during solid state isothermal ageing was controlled by the bulk diffusion process. The time exponent n of non- reinforced SAC305 was slightly higher than SAC305-KGC composite solder with an increase in temperature. This phenomenon can be associated with the changes in the morphology of IMC layer. As discussed in Section 3.1, at 150 °C, the SAC305-KGC composite solder had more planar-type morphology in which the channel between the scallop gradually vanishes. Thus, the diffusion process took shorter time and led to small value of time exponent n. This result was aligned with the research by Tang et al. [11]. Meanwhile, for Cu_3_Sn IMC layer, the value of time exponent n was in the range of 0.21–0.29. From the values obtained, the growth of Cu_3_Sn layer was controlled by grain-boundary diffusion process. Based on the results attained, it was noticed the total interfacial IMC and Cu_3_Sn exhibited different IMC growth controlling mechanism. This was due to the difference in the microstructure formation and thickness during the isothermal ageing process [32].

Moreover, Figure 14 presents the relationship between total interfacial IMC thickness layer (including Cu_6_Sn_5_ and Cu_3_Sn) and Cu_3_Sn IMC with several ageing temperatures and times for non-reinforced SAC305 and SAC305-KGC composite solders. Based on the graph plotted in Figure 14, the total interfacial IMC thickness layer and Cu_3_Sn IMC increased with an increase ageing time and grew faster at higher ageing temperature. Interestingly, the thickness of IMC layer was suppressed with the incorporation of KGC during the solid-state ageing. Quantitative investigation on the effect of KGC particles towards the interfacial thickness of IMC layer, across non-reinforced SAC305 and SAC305-KGC composite solders during the solid-state ageing, can be described according to Equation (1). The diffusion coefficient D can be determined from a linear regression analysis by plotting the graph presented in Figure 14, where the slope of the graph is equal to D. Thus, the diffusion coefficient, D for the growth of total interfacial IMC layer and Cu_3_Sn IMC in non-reinforced SAC and SAC305-KGC composite solders are presented in Table 1. Based on the values of diffusion coefficient obtained, SAC305-KGC composite solder showed lower D value compared to SAC solder at all ageing temperatures. This can be inferred that, KGC particles could hinder the growth of IMC between the solder and substrate. Additionally, at a temperature of 150 °C with ageing time of 1000 h, the values of D for both solders increased corresponding to the thicker IMC at the interfacial layer. This owes to the fact that higher ageing temperature can provide appropriate thermal energy in order to overcome the higher activation energy of diffusion elements [27]. The results obtained was parallel with the studied done by Yin et al. [33].

Meanwhile, the activation energy was determined according to the Arrhenius equation expressed in Equation (3):(3)$${D=\mathrm{Doe}}^{-Q/\mathrm{RT}}$$
where Do is the temperature-dependent constant, Q is activation energy (kJ/mol), R is universal gas constant and T is the absolute temperature in degrees Kelvin. The activation energy of the interfacial intermetallic compound can be calculated by taking natural logarithm of Equation (3). Thus, the diffusion coefficient D can be expressed as:(4)$$\ln D=\ln\mathrm{Do}-\frac{Q}{R}\;\left(\frac{1}{T}\right)$$

Equation (4) takes the form of y = mx + C, where the independent variable is (1/T) and dependent variable is ln D. By using Equation (4), the activation energy (Q) can be calculated by plotting the graph of ln D vs. 1/T where the slope of the graph represents the value of Q by using linear regression model. Based on the Equation (4) and the Arrhenius plotted in Figure 15, the activation energy for the total interfacial IMC layer and Cu_3_Sn layer can be calculated. 

The activation energy for the total interfacial IMC layer in SAC305 solder was 52 kJ/mol, while, for SAC305-KGC composite solder, the activation energy was 74 kJ/mol. As reported by [34,35,36], the activation energy for SAC solder alloy was in the range of ~44 to 77 kJ/mol. Therefore, the activation energy obtained from this research was comparable with the previous studies. In addition to this, the higher activation energy value in SAC305-KGC composite solder corresponds to the low growth rate at lower temperature and higher growth rate at high temperature, as the value of activation energy was obtained from temperature dependence of diffusion coefficient, D. The higher activation energy experienced by SAC305-KGC composite solders was parallel with the lower total interfacial thickness of IMC layer. Furthermore, the activation energy of Cu_3_Sn layer (as plotted in Figure 15b), was 98 kJ/mol and 104 kJ/mol for SAC305 and SAC305-KGC composite solders, respectively. This result indicates that the growth of Cu_3_Sn layer in SAC305-KGC composite solder was slightly slower than SAC305 solder. As discussed earlier, the slightly lower activation energy exhibited by SAC305-KGC solders might be due to the effects of KGC particles addition in the solder. Moreover, in a study done by Tang et al. [11], they reported that TiO_2_ particles display a little effect in increasing the activation energy of Cu_3_Sn layer for Sn-3.0Ag-0.5Cu lead-free solder.

### 3.3. Shear Joint Strength

Figure 16 depicts the results on average shear strength of non-reinforced SAC305 and SAC305-KGC composite solders prior to reflow soldering and subjected to isothermal ageing of 150 °C at 100 h, 500 h and 1000 h. The average shear strength of non-reinforced SAC305 and SAC305-KGC composite solders after reflow soldering were ~10 MPa and ~13 MPa, respectively. The average shear strength of SAC305-KGC composite solder showed 31% improvement compared to non-reinforced SAC305 after reflow soldering process. However, during the isothermal ageing treatment at higher temperature of 150 °C for 100 h, 500 h and 1000 h, the average shear strength for non-reinforced SAC305 and SAC305-KGC composite solders displayed declining trend. The average shear strength decreased with the increase in temperature and longer ageing time. Nevertheless, the average shear strength of SAC305-KGC composite solder was still higher across all the samples aged at different time and temperature as compared to non-reinforced SAC305 solders. The lowest average shear strength was shown by non-reinforced SAC305 solder aged at 150 °C for 1000 h, which is about ~5 MPa. It was expected that, at higher ageing time and temperature, the shear strength is poor. This can be associated with the increment in the total thickness of interfacial IMC layer as depicted in Figure 11. The decreasing trend of the average shear strength across all the samples were corresponding to the thickness of the IMC layer. The thicker the IMC layer, the lower the average shear strength. This is because, IMC is brittle. Thus, thicker IMC layers were easily exposed to brittle failure and consequently, reducing the strength of solder joints. Nevertheless, the average shear strength in SAC305-KGC composite solder was still higher than non-reinforced SAC305 solders. This suggests that KGC particles play an important role in improving the shear strength during ageing process. In addition to this, KGC particles are also believed to be able to pin the dislocation motions and hinder the grain-boundary sliding across the solder matrix. The results obtained was in good agreement with the study done by Chen et al. [8]. He reported that the improvement in the shear strength of Sn-3.0Ag-0.5Cu during solid-state ageing was attributed to the theory of dispersion strengthening resulting from the addition of reinforcement particles.

To ascertain the fracture mode for the samples after the lap shear test, the samples were observed under a scanning electron microscope (SEM) in secondary electron (SEI) mode as presented in Figure 17. Figure 17a,e represent the fracture surfaces of non-reinforced SAC305 and SAC305-KGC composite solder prior to reflow soldering, respectively. Based on the figure it can be clearly observed that during reflowed condition, non-reinforced SAC305 exhibited a combination of fracture mode which is between ductile and brittle mode, whereas for SAC305-KGC sample, some dimples could be observed which correspond to the ductile fracture mode. By increasing the isothermal ageing temperature and time up to 150 °C for 1000 h, the fracture mode of the samples were slightly changed. Evidently, in SAC305-KGC composite solder, ductile dimple morphology was observed at the samples aged for 100 h and 500 h (as in Figure 17f,g). In addition, the dimple size areas were increased with the increase in ageing time which can be attributed to the coarse planar IMC layer, as suggested by Dele-Afolabi et al. [37]. Additionally, ductile brittle fracture mode was dominant in non-reinforced SAC305 aged at 150 °C for 1000 h, as presented in Figure 17g. The brittle fracture occurred in the region of intermetallics interface [38]. At higher ageing condition, non-reinforced SAC305 had a thicker IMC layer which supports the reason for the occurrence of the brittle fracture mode shown by the sample. Nevertheless, in SAC305-KGC composite solder, the fracture mode was a combination between ductile and brittle fracture modes. The improvement in the fracture mode of SAC305 was partially attributed to suppression of IMC layer in the samples with KGC particles. From this, it can be concluded that the KGC particles are able to alter the fracture mode even though the samples were subjected to higher isothermal ageing temperature for a longer time.

## 4. Conclusions

The effects of KGC particles as the reinforcement in SAC305 solder had been investigated at temperature of 85 °C, 125 °C and 150 °C for 100 h, 500 h and 1000 h. The following conclusions can be gathered:(a)The morphology of interfacial IMC layer of non-reinforced SAC305 and SAC305-KGC composite solder joints showed a duplex IMC structure comprises of scallop-type Cu_6_Sn_5_ and layer-type Cu_3_Sn. With an increase in ageing time and temperature, the initial scallop Cu_6_Sn_5_ gradually changes to planar-type in SAC305-KGC composite solder joints. For Cu_3_Sn, the morphology consistently maintained as layer-type across all ageing conditions.(b)The total thickness of interfacial IMC (both Cu_6_Sn_5_ and Cu_3_Sn) layer showed an increasing trend for non-reinforced SAC305 and SAC305-KGC composite solder joints which was about ~6–17 µm. However, with an addition of KGC particles the total thickness of interfacial IMC could be suppressed for about 15% as compared to non-reinforced SAC305 solder joints with the increase in ageing time and temperature. This can be owed to the ability of KGC particles to hinder the diffusion of Cu and resulting in thinner IMC layer. Moreover, the growth of total interfacial IMC layer was controlled by bulk diffusion process with the time exponent, n obtained was towards 0.5. The activation energy for SAC305-KGC composite solder joints was 74 kJ/mol and it exhibited lower diffusion coefficient as compared to non-reinforced SAC305.(c)Meanwhile, the addition of KGC particles into SAC305 could suppress the growth of Cu_3_Sn IMC layer by 24% as compared to non-reinforced SAC305 solder. In addition, the growth of Cu_3_Sn IMC layer in this study was controlled by grain-boundary diffusion. The activation energy of Cu_3_Sn IMC layer for SAC305-KGC composite solder joints was 104 kJ/mol and it exhibited lower diffusion coefficient even at high temperature as compared to non-reinforced SAC305.(d)The average shear strength for all the solder joints decreased with the increase ageing time and temperature. However, the decrement of the strength was lower in SAC305-KGC composite solder joints than non-reinforced SAC305 solder. The average shear strength of SAC305-KGC composite solder was in the range of ca. 8–13 MPa. Meanwhile for non-reinforced SAC305, the average shear strength was in the range of ca. 5–9 MPa. In addition, SAC305-KGC solder joint possessed a combination of ductile and brittle fracture mode at higher temperature of 150 °C and 1000 h of ageing time.

## Figures and Tables

**Figure 1 materials-14-00776-f001:**
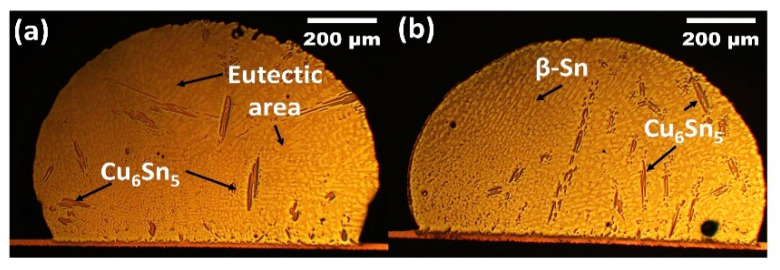
Microstructure of as-reflowed solder bulk for (**a**) non-reinforced SAC305 and (**b**) SAC305-KGC composite solders.

**Figure 2 materials-14-00776-f002:**
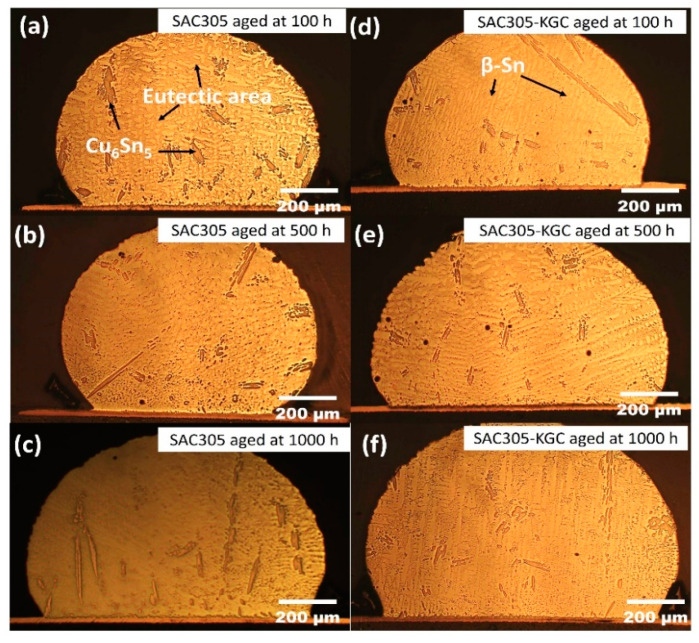
Microstructure formation at the bulk solder after isothermal ageing at 85 °C. Non-reinforced SAC305 solder at (**a**) 100 h, (**b**) 500 h, (**c**) 1000 h and SAC305-KGC composite solder at (**d**) 100 h, (**e**) 500 h, (**f**) 1000 h.

**Figure 3 materials-14-00776-f003:**
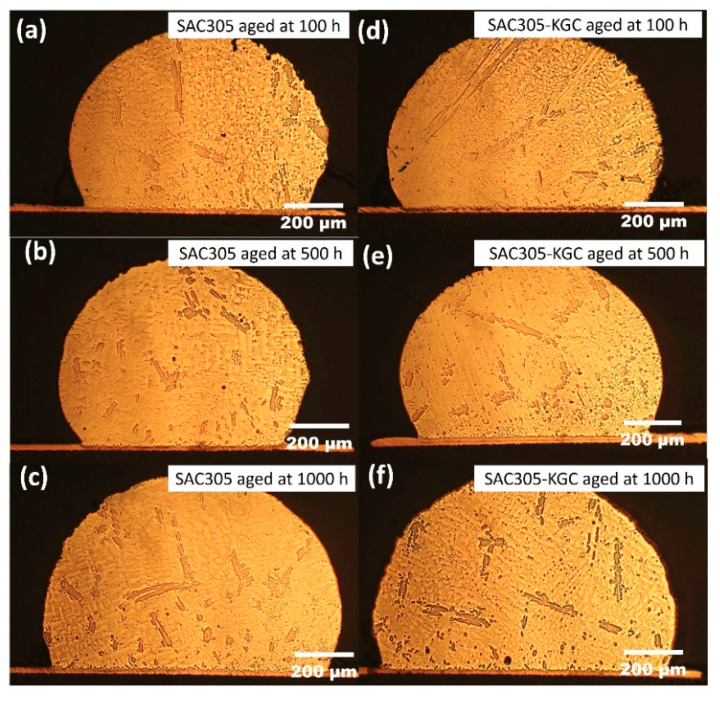
Microstructure formation at the bulk solder after isothermal ageing at 125 °C. Non-reinforced SAC305 solder at (**a**) 100 h, (**b**) 500 h, (**c**) 1000 h and SAC305-KGC composite solder at (**d**) 100 h, (**e**) 500 h, (**f**) 1000 h.

**Figure 4 materials-14-00776-f004:**
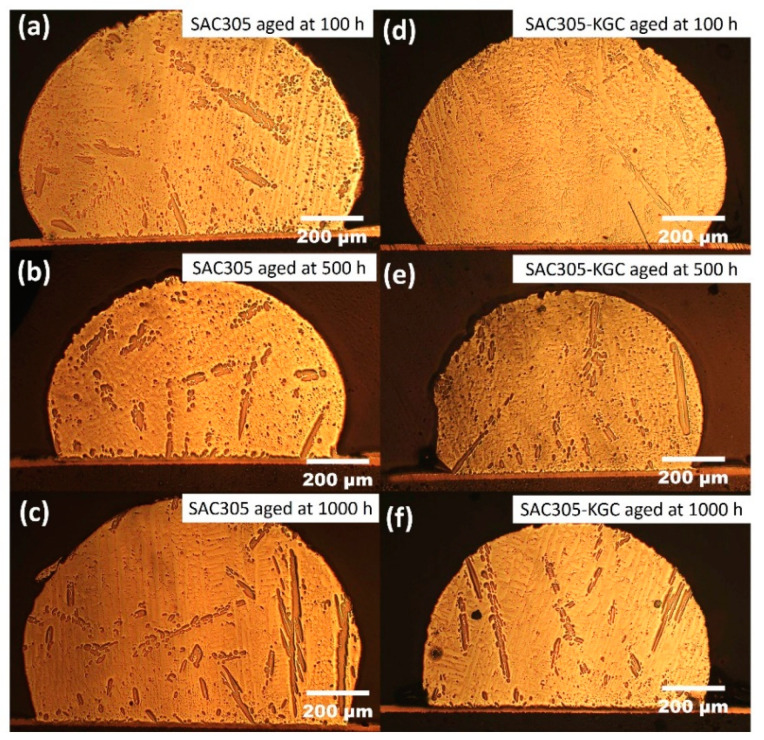
Microstructure formation at the bulk solder after isothermal ageing at 150 °C. Non-reinforced SAC305 solder at (**a**) 100 h, (**b**) 500 h, (**c**) 1000 h and SAC305-KGC composite solder at (**d**) 100 h, (**e**) 500 h, (**f**) 1000 h.

**Figure 5 materials-14-00776-f005:**
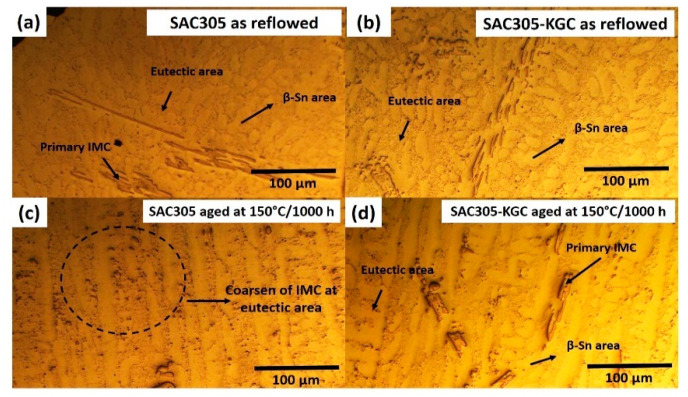
Microstructure at the bulk solder for (**a**) non-reinforced SAC305 as reflowed, (**b**) SAC305-KGC as reflowed, (**c**) SAC305 aged at 150 °C for 1000 h and (**d**) SAC305-KGC aged at 150 °C for 1000 h.

**Figure 6 materials-14-00776-f006:**
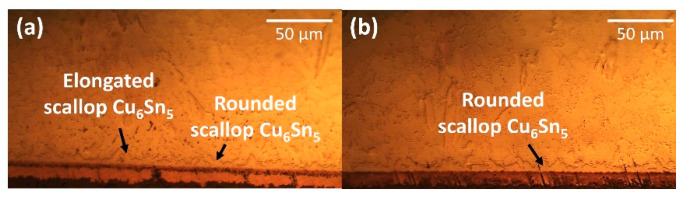
Cross-sectional as-reflowed optical microscope images for (**a**) non-reinforced SAC305 and (**b**) SAC305-KGC composite solder.

**Figure 7 materials-14-00776-f007:**
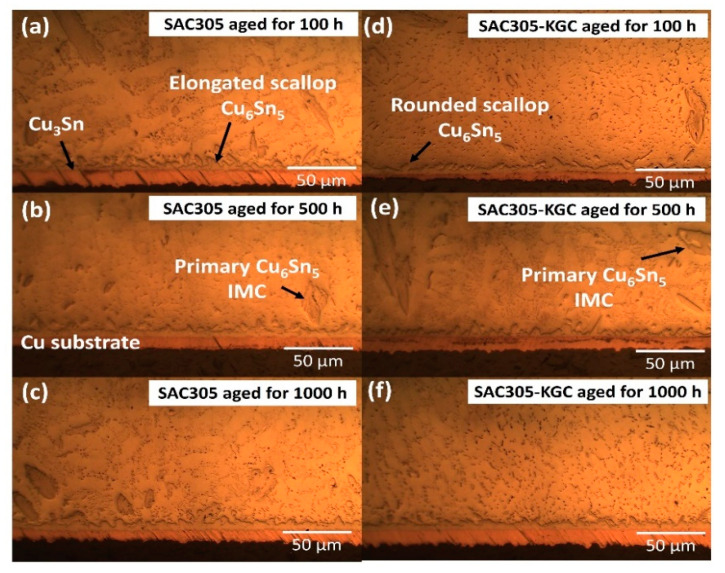
Interfacial IMC layer subjected to isothermal ageing at 85 °C. Non-reinforced SAC305 at (**a**) 100 h, (**b**) 500 h, (**c**) 1000 h. SAC305-KGC composite solder at (**d**) 100 h, (**e**) 500 h and (**f**) 1000 h.

**Figure 8 materials-14-00776-f008:**
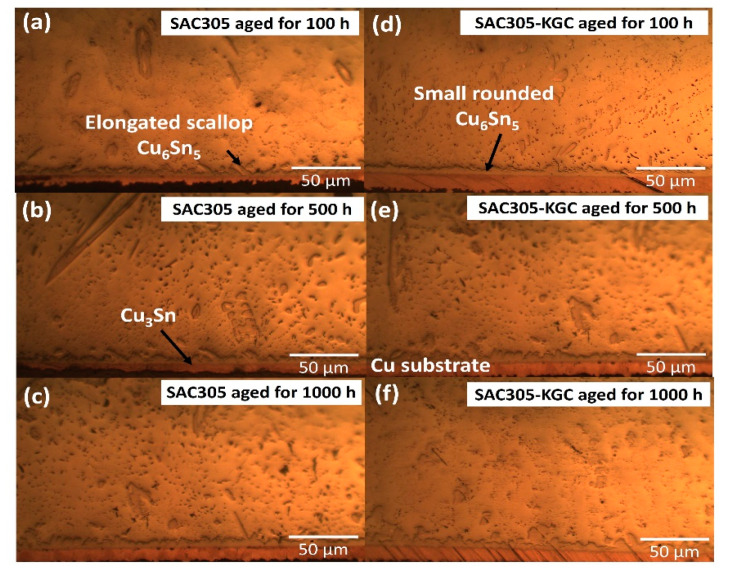
Interfacial IMC layer subjected to isothermal ageing at 125 °C. Non-reinforced SAC305 at (**a**) 100 h, (**b**) 500 h, (**c**) 1000 h. SAC305-KGC composite solder at (**d**) 100 h, (**e**) 500 h and (**f**) 1000 h.

**Figure 9 materials-14-00776-f009:**
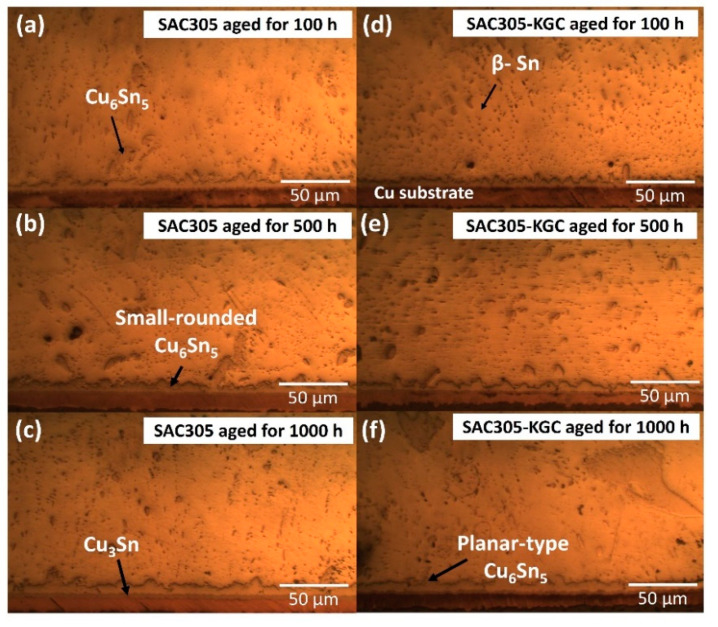
Interfacial IMC layer subjected to isothermal ageing at 150 °C. Non-reinforced SAC305 at (**a**) 100 h, (**b**) 500 h, (**c**) 1000 h. SAC305-KGC composite solder at (**d**) 100 h, (**e**) 500 h and (**f**) 1000 h.

**Figure 10 materials-14-00776-f010:**
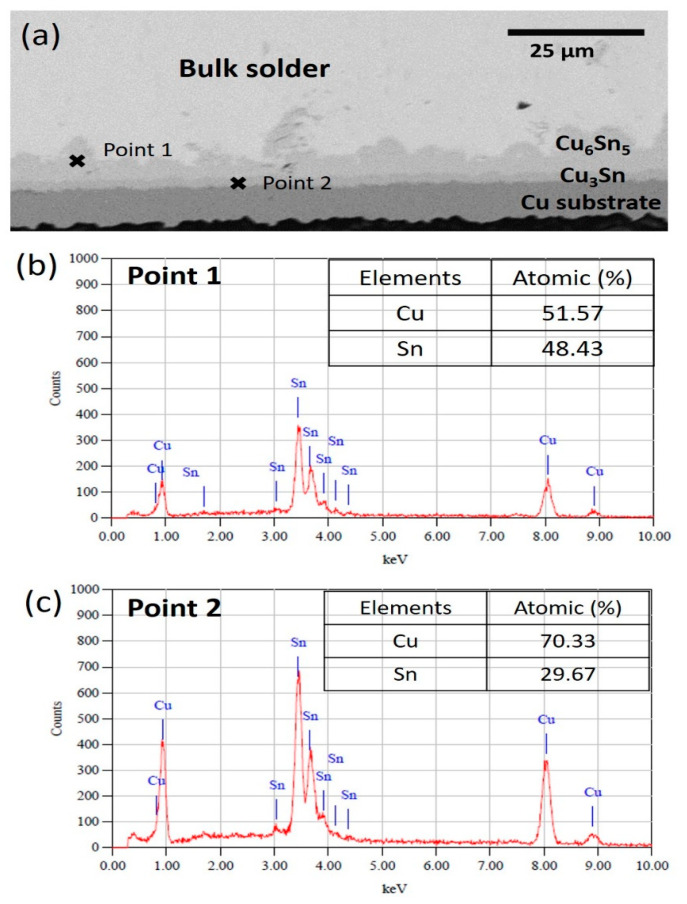
(**a**) Cross-sectional images of EDX point analysis. (**b**) EDX analysis result at point 1 and (**c**) EDX analysis result at point 2.

**Figure 11 materials-14-00776-f011:**
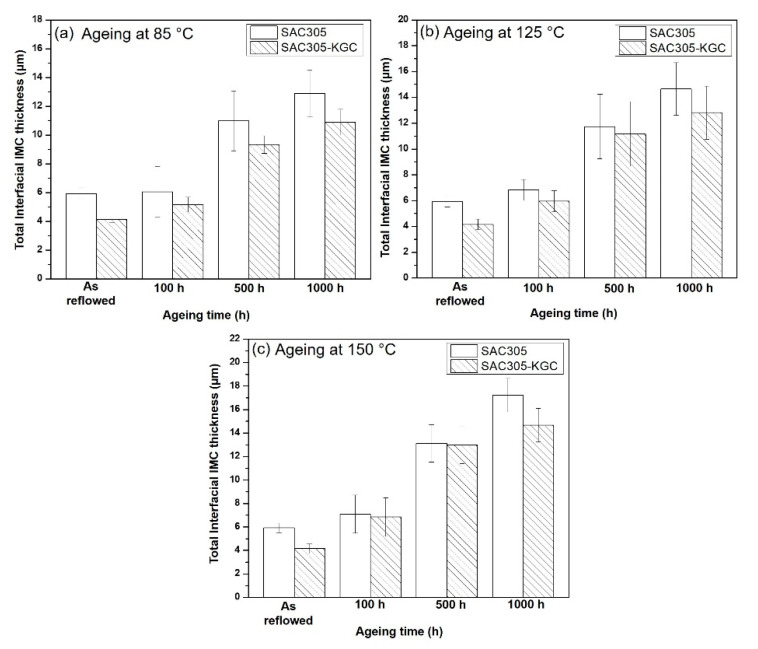
Total interfacial IMC thickness subjected to isothermal ageing at temperature of (**a**) 85 °C, (**b**) 125 °C and (**c**) 150 °C.

**Figure 12 materials-14-00776-f012:**
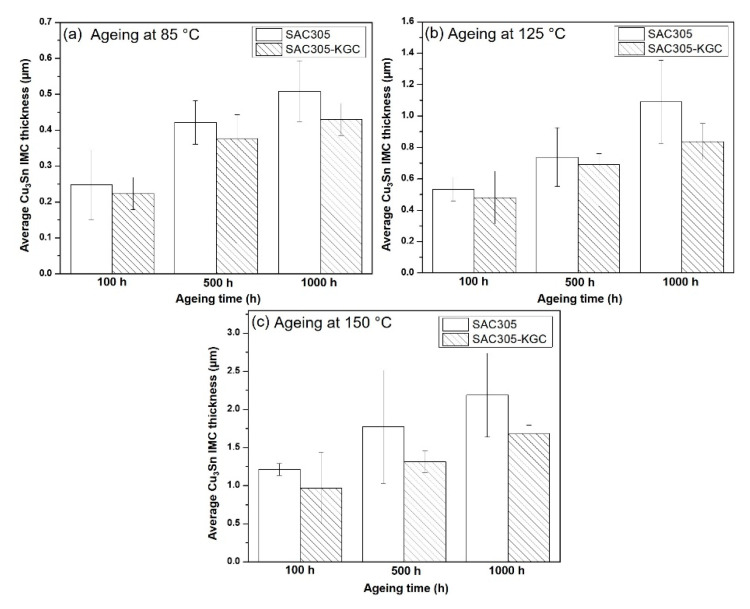
Average Cu_3_Sn IMC thickness subjected to isothermal ageing at temperature of (**a**) 85 °C, (**b**) 125 °C and (**c**) 150 °C.

**Figure 13 materials-14-00776-f013:**
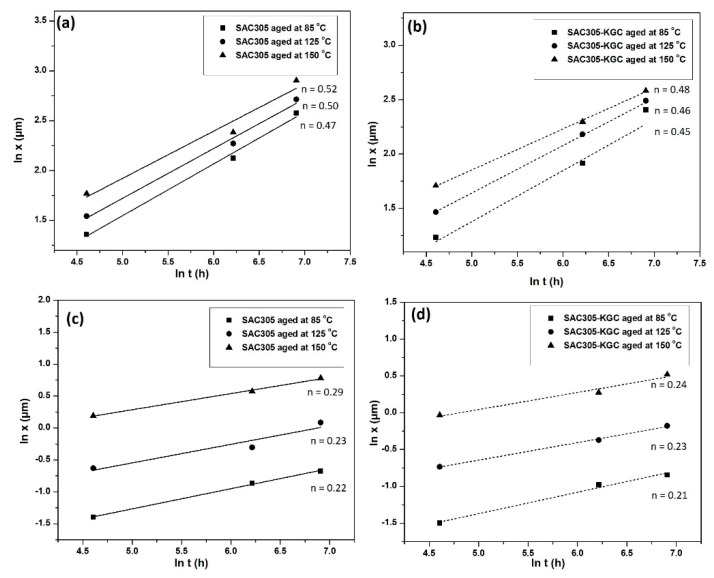
Time exponent of Cu_6_Sn_5_ for (**a**) non-reinforced SAC305, (**b**) SAC305-KGC composite solder. Cu_3_Sn, (**c**) non-reinforced SAC305 and (**d**) SAC305-KGC composite solder.

**Figure 14 materials-14-00776-f014:**
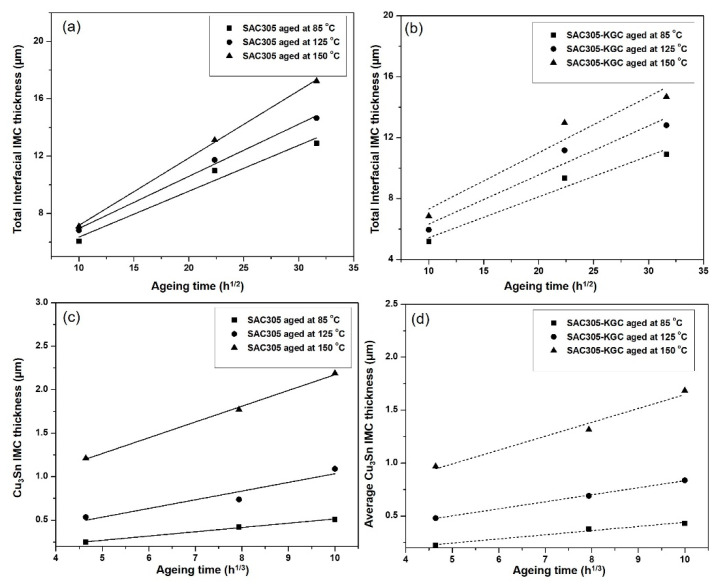
The relationship between the total interfacial IMC thickness curves and ageing temperature and ageing time for (**a**) total interfacial IMC for non-reinforced SAC305, (**b**) total interfacial IMC for SAC305-KGC composite solder (**c**) Cu_3_Sn IMC for non-reinforced SAC305 and (**d**) Cu_3_Sn IMC for SAC305-KGC composite solder.

**Figure 15 materials-14-00776-f015:**
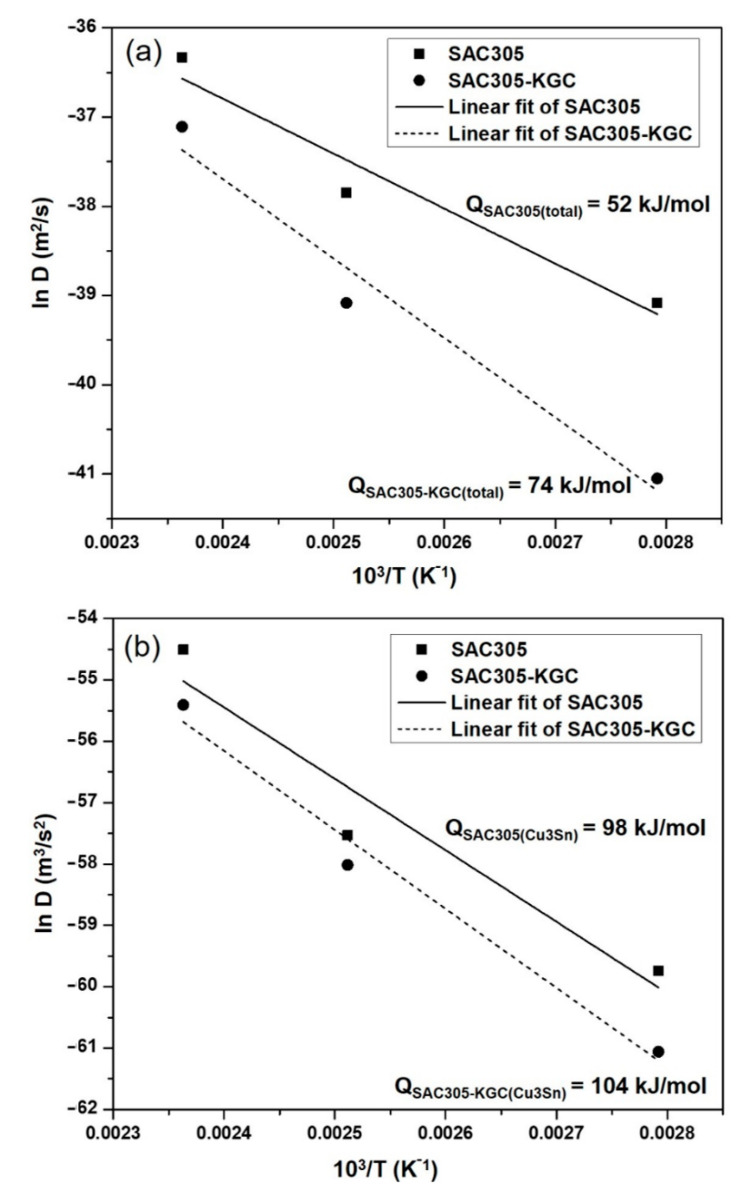
Arrhenius plot of ln D vs. 1/T for (**a**) total interfacial IMC layer and (**b**) Cu_3_Sn layer in SAC305 and SAC305-KGC composite solder.

**Figure 16 materials-14-00776-f016:**
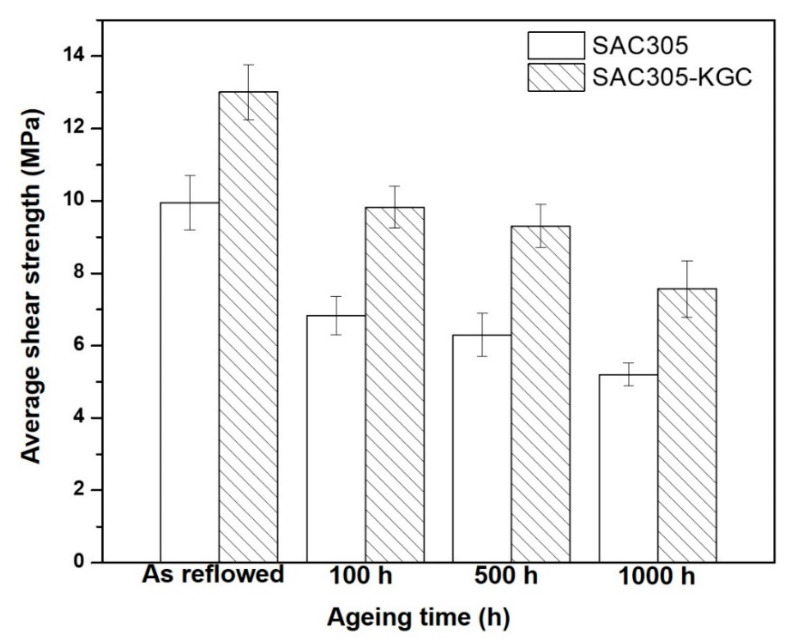
Average shear strength of non-reinforced SAC305 and SAC305-KGC composite prior to reflow soldering and subjected to isothermal ageing at 150 °C.

**Figure 17 materials-14-00776-f017:**
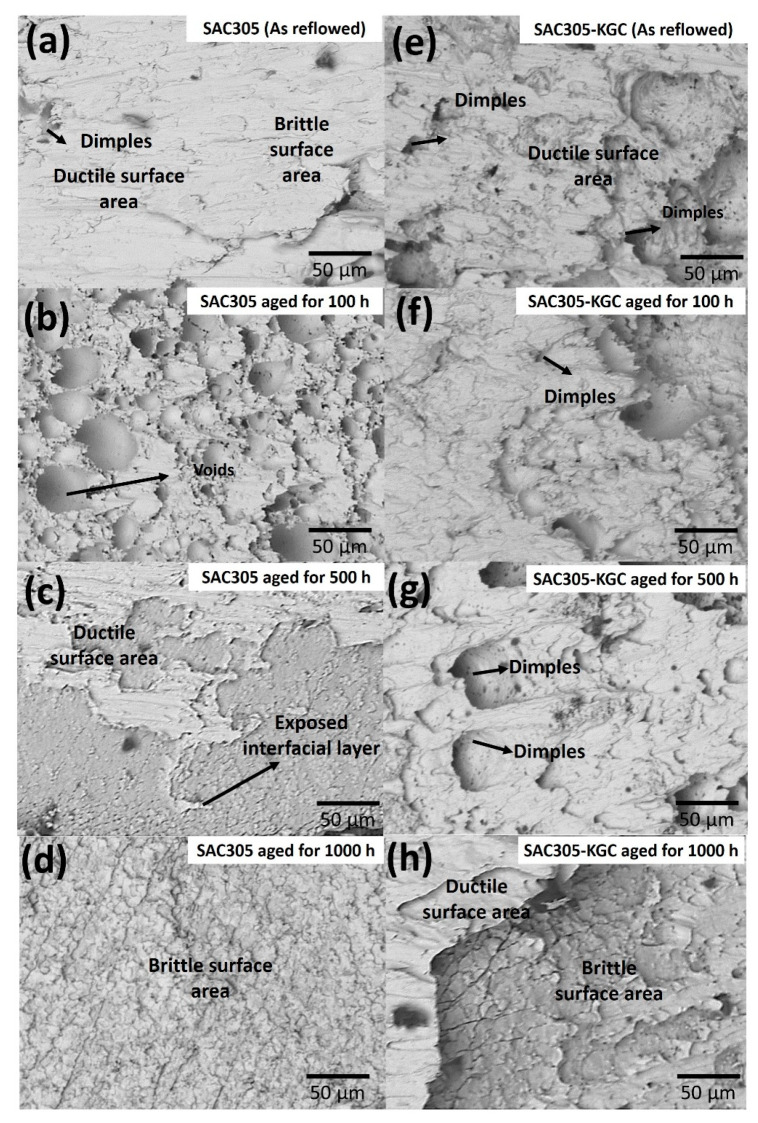
SEM micrographs of fracture surfaces subjected to isothermal ageing at 150 °C for non-reinforced SAC305 (**a**) as-reflowed, (**b**) 100 h, (**c**) 500 h, (**d**) 1000 h and SAC305-KGC composite solder (**e**) as-reflowed, (**f**) 100 h, (**g**) 500 h and (**h**) 1000 h.

**Table 1 materials-14-00776-t001:** Diffusion coefficient for total interfacial IMC layer and Cu_3_Sn IMC.

Solder Composition	Temperature (°C)	Diffusion Coefficient (Total Interfacial IMC)	Diffusion Coefficient (Cu_3_Sn IMC)
Non-reinforced SAC305	85	2.89 × 10^−17^	3.29 × 10^−26^
	125	3.66 × 10^−17^	2.77 × 10^−25^
	150	4.25 × 10^−17^	1.65 × 10^−24^
SAC305-KGC composite solder	85	1.84 × 10^−17^	1.68 × 10^−26^
	125	2.89 × 10^−17^	8.09 ×10^−26^
	150	3.79 × 10^−17^	6.24 × 10^−25^

## Data Availability

The data presented in this study are available on request from the corresponding author.

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
