# Peer review of "Performance of Sn-3.0Ag-0.5Cu Composite Solder with Kaolin Geopolymer Ceramic Reinforcement on Microstructure and Mechanical Properties under Isothermal Ageing"

_materials, 2021, doi:10.3390/ma14040776_

Round 1

Reviewer 1 Report

The article is interesting. Overall, it is well written.
But you can add the following to the review as questions or clarifications:
1.What is the reason for the fact that the Cu3Sn phase is not observed in all states?
2. The Cu6Sn5 Phase have different shapes in different states. What is the reason for this difference?
3. What is the role of particles Cu3Sn and Cu6Sn5 phases in mechanical behavior?
4. What is the reason for the slight difference in Average shear strength at aging of 100 and 500 hours?
5.Did you study the microstructure by transmission electron microscopy or X-ray structural analysis? For fine particle analysis.

Author Response

Manuscript ID : materials-1092704

Title: “Performance of Sn-3.0Ag-0.5Cu composite solder with kaolin geopolymer
ceramic reinforcement on microstructure and mechanical properties under
isothermal ageing”

Dear Editor,                                                                                                      

Please find the reviewer’s comments and corrections been made by authors as below:

REVIEWER 1

No

Comment

Corrections

1.

What is the reason for the fact that the Cu3Sn phase is not observed in all states?

Thank you for the comments. In the manuscript (line 197), it was mentioned that Cu3Sn phase was not observed only in the samples of as-reflowed for both non-reinforced SAC305 and SAC305-KGC composite solders. Using our analysis technique (optical microscope and scanning electron microscopy), the Cu3Sn layer was not observable where it could be detected using higher resolution microscopes such as Transmission Electron Microscopy (TEM). In our study, the Cu3Sn phase was observed in all samples after isothermal ageing as revealed in Figure 7, Figure 8, and Figure 9 in the manuscript.

2.

The Cu6Sn5 Phase have different shapes in different states. What is the reason for this difference?

Thank you for the comments. The reaction of Sn atoms from the solder and Cu atoms from the substrate (Cu diffusion) during soldering and ageing will influence the morphology of the interfacial Cu6Sn5. Specifically, as increasing ageing time and temperature, the interfacial Cu6Sn5 become flatter to a more planar type. This can be described according to the figure below:

The distance between the scallop valleys (dv) and the Cu substrate was closer as compared to the distance between scallop peaks (dp) and the Cu substrate. Therefore, the Sn atoms from the solder and Cu atoms from substrate react to each other faster than other regions, thus lead to higher growth rate at the valley regions and changed the morphology of interfacial Cu6Sn5. The changes of interfacial Cu6Sn5 morphology were widely discussed as reported below:

1. Wang, F., et al., Microstructural evolution and joint strength of Sn-58Bi/Cu joints through minor Zn alloying substrate during isothermal aging. Journal of Alloys and Compounds, 2016. 688: p. 639-648.

2. Tang, Y., et al., Effect of Nano-TiO2 particles on growth of interfacial Cu6Sn5 and Cu3Sn layers in Sn–3.0Ag–0.5Cu–xTiO2 solder joints. Journal of Alloys and Compounds, 2016. 684.

3. Gain, A.K. and L. Zhang, Nanosized samarium oxide (Sm2O3) particles suppressed the IMC phases and enhanced the shear strength of environmental-friendly Sn–Ag–Cu material. Materials Research Express, 2019. 6(6): p. 066526.

4. Feng, J., et al., Growth kinetics of Cu6Sn5 intermetallic compound in Cu-liquid Sn interfacial reaction enhanced by electric current. Scientific Reports, 2018. 8.

Besides that, the changes in the morphology also relates with the different in the Gibs Free energy. The changes in the IMC structures during the isothermal ageing was to lower the surface energy. The surface energy of scallop IMC was higher than the layered type. The heat produce from the isothermal ageing process causes the surface tension of the Cu6Sn5 to be unstable. To stabilize the surface tension, the excess energy was removed through atoms diffusion resulting in higher Cu atom diffusion thus produced layered type of Cu6Sn5 IMC.

This explanation has also been added in the manuscript (Line 224-231).

3.

What is the role of particles Cu3Sn and Cu6Sn5 phases in mechanical behavior?

Cu6Sn5 and Cu3Sn are common interfacial IMCs that commonly forms in Sn-Cu solder joints. The interfacial IMC forms as a result of Sn and Cu atom diffusion during soldering and it is also an indication of a good wetting for soldering. Both intermetallic (Cu6Sn5 and Cu3Sn) are brittle in nature where cracks may initiate. Thus, formation of large and thick layer of Cu3Sn and Cu6Sn with increasing aging time and temperature may reduce the mechanical performance of the solder.

4.

What is the reason for the slight difference in Average shear strength at aging of 100 and 500 hours?

The slight difference in the average shear strength at aging of 100 and 500 hours may due to the several factors. As reported by [M. A. A. Mohd Salleh, S. D. McDonald, and K. Nogita, "Effects of Ni and TiO2 additions in as-reflowed and annealed Sn0.7Cu solders on Cu substrates," Journal of Materials Processing Technology, vol. 242, pp. 235-245, 2017/04/01/ 2017], the thickness of interfacial IMC layer was not only the factor that contributes to the shear strength. However, other factors may also be considered such as the solder flux void formation, Kirkendal void formation and large primary IMCs. Thus in this study, on of the factors that might influence the slight difference in the shear strength may be attributed to the flux void formation. As according to the failure mode observed under scanning electron microscope [Figure 17(b) in the manuscript], there are many voids which could reduce the strength of the solder even at low ageing time (100 hours).

5.

Did you study the microstructure by transmission electron microscopy or X-ray structural analysis? For fine particle analysis.

In this research, we only used optical microscope and scanning electron microscope (coupled with EDS) to study the microstructure formation. We believe that the use of these technique could give sufficient information on the topic covered in this manuscript. However, we may use the transmission electron microscopy or x-ray structural analysis to investigate more on the structure and crystallography of the phases found in our solder joint. Thank you for your suggestion.

We thank you for the valuable suggestions made by the reviewers to improve our paper. All concerns and suggestions have been addressed in the above feedback. We would like to thank the editor in advance for considering our work.

Yours sincerely,

Dr Mohd Arif Anuar Mohd Salleh

Nihon Superior Electronic Material Research Lab, Center of Excellence Geopolymer & Green Technology (CeGeoGTech), Faculty of Chemical Engineering Technology, Universiti Malaysia Perlis (UniMAP), Taman Muhibbah, 02600, Arau, Perlis, Malaysia.

Reviewer 2 Report

It is very nicely written paper. It is a useful topic, can be relevant for innovations in industry.

Basically it could be accepted as prepared but there are some technical issues that should be taken care of:

  1. line 63: "have been chosen as one of the materials" -> "have been chosen as the ones"
  2. There are some problems on page 7 - Figure 7 is repeated several times across the text, there are some format problems. Also the sentence in line 212 is somehow truncated - which is probably connected to the issue with Fig. 7.
  3. The explanation of the linearization and evaluation of the Eq (3) on the lines 364 to 372 (including Eq(7)) is completely unnecessary. This is a very trivial process clear for every reader - I hope.

Apart of those, I have a question, that possibly could be commented:

  1. The study mentions only the effect of cracking and other problems on mechanical properties. Do the authors have any concerns or did they performed any study of changes of electrical conductivity of the solder connected to cacking and/or new phases?

After addressing these issues, the paper can be published.

Author Response

Manuscript ID : materials-1092704

Title: “Performance of Sn-3.0Ag-0.5Cu composite solder with kaolin geopolymer
ceramic reinforcement on microstructure and mechanical properties under
isothermal ageing”

Dear Editor,                                                                                                      

Please find the reviewer’s comments and corrections been made by authors as below:

REVIEWER 2

No

Comment

Correction

1.

line 63: "have been chosen as one of the materials" -> "have been chosen as the ones"

Thank you for your comments. We have changed "have been chosen as one of the materials" to “have been chosen as the ones" in the manuscript.

2.

There are some problems on page 7 - Figure 7 is repeated several times across the text, there are some format problems. Also the sentence in line 212 is somehow truncated - which is probably connected to the issue with Fig. 7.

Thank you for your comments. We have double checked the format and it is now in a correct version. We have also reduced the unnecessary repetition in our manuscript.

3.

The explanation of the linearization and evaluation of the Eq (3) on the lines 364 to 372 (including Eq(7)) is completely unnecessary. This is a very trivial process clear for every reader - I hope.

Thank you for your comment. We believe that the explanation of the linearization and evaluation of Eq.3 may help the readers to clearly understand on how the growth kinetics of IMC were calculated.

4.

The study mentions only the effect of cracking and other problems on mechanical properties. Do the authors have any concerns or did they performed any study of changes of electrical conductivity of the solder connected to cacking and/or new phases?

Thank you for the comment. In this study we only focused on the effects of addition Kaolin geopolymer ceramics to SAC305 lead free solder subjected to isothermal ageing process where majorly the microstructure study was performed and correlated with the mechanical performance. In this manuscript we did not performed any study related to the electrical conductivity. However, it is a good suggestion for us to explore in future.

We thank you for the valuable suggestions made by the reviewers to improve our paper. All concerns and suggestions have been addressed in the above feedback. We would like to thank the editor in advance for considering our work.

Yours sincerely,

Dr Mohd Arif Anuar Mohd Salleh

Nihon Superior Electronic Material Research Lab, Center of Excellence Geopolymer & Green Technology (CeGeoGTech), Faculty of Chemical Engineering Technology, Universiti Malaysia Perlis (UniMAP), Taman Muhibbah, 02600, Arau, Perlis, Malaysia.

Reviewer 3 Report

It is a very interesting contribution investigatin the effect of the addition of kaolin in a composite solder. The article is well organized and the conclusions well supported. Some minor issues should be addressed before publication.

1. It is nice that the related references are new, but SAC solders and in general lead-free ones have been studied since several years and acknowledgement to pioneering works has to be paid. Please introduce some of them or reviews among the references for the first part.

For example: Fix, A.R., López, G.A., Brauer, I., Nüchter, W., Mittemeijer, E.J. Microstructural development of Sn-Ag-Cu solder joints. Journal of Electronic Materials 34(2) (2005) pp. 137-142

Furthermore, concerning the growth kinetics observed of Cu-Sn intermetallics there are a lot of previous works which should cited (Look in database for professors from many different reknown centers:E.J. Mittemeijer, W. Gust, P. Zieba, J.M. Johnson, R. Fournelle, L.C. Tsao, etc.)

For example:

Dariavach, N., Callahan, P., Liang, J., Fournelle, R. Intermetallic growth kinetics for Sn-Ag, Sn-Cu, and Sn-Ag-Cu lead-free solders on Cu, Ni, and Fe-42Ni substrates. Journal of Electronic Materials 35(7) (2006) 1581-1592

2. As usual, EDS results are reported with an extremely low uncertainty. This is not true. Standarless EDS analyses can never have an error smaller than 5%. Therefore, please correct table in Figs. 10b and 10c. It not critical for the conclusions of this article, but must be corrected.

3. What do the error bars given in Figs. 11 and 12 stand for? Explain in the text.

4. Just a recommendation for the future. To make linear regression with only three points might be acceptable, but introducing more points is adviceble. 

Author Response

Manuscript ID : materials-1092704

Title: “Performance of Sn-3.0Ag-0.5Cu composite solder with kaolin geopolymer
ceramic reinforcement on microstructure and mechanical properties under
isothermal ageing”

Dear Editor,                                                                                                      

Please find the reviewer’s comments and corrections been made by authors as below:

REVIEWER 3

No

Comment

Corrections

1.

It is nice that the related references are new, but SAC solders and in general lead-free ones have been studied since several years and acknowledgement to pioneering works has to be paid. Please introduce some of them or reviews among the references for the first part.

Thank you for comments and suggestions. We have added the suggested references in the manuscript.

2.

As usual, EDS results are reported with an extremely low uncertainty. This is not true. Standarless EDS analyses can never have an error smaller than 5%. Therefore, please correct table in Figs. 10b and 10c. It not critical for the conclusions of this article, but must be corrected.

Yes, we agree that the accuracy of EDS is not that high compared to a more precise technique. However, in our study, EDS were used to show a pre-indication of which phase is present where common phases in Sn-Cu solder joints were known to form. Similarly reported by other studies below, EDS may be used to pre-indicate which phase is present:

1. X. Hu, T. Xu, L. M. Keer, Y. Li, and X. Jiang, "Microstructure evolution and shear fracture behavior of aged Sn3Ag0.5Cu/Cu solder joints," Materials Science and Engineering: A, vol. 673, pp. 167-177, 2016/09/15/ 2016.

2. L. Liu, Z. Chen, C. Liu, Y. Wu, and B. An, "Micro-mechanical and fracture characteristics of Cu6Sn5 and Cu3Sn intermetallic compounds under micro-cantilever bending," Intermetallics, vol. 76, pp. 10-17, 2016/09/01/ 2016.

3. N. Jiang, L. Zhang, W.-m. Long, S.-j. Zhong, and L. Zhang, "Influence of doping Ti particles on intermetallic compounds growth at Sn58Bi/Cu interface during solid–liquid diffusion," Journal of Materials Science: Materials in Electronics, 2021/01/19 2021.

4. H. Wang, X. Hu, Q. Li, and M. Qu, "Effect of flux doped with Cu6Sn5 nanoparticles on the interfacial reaction of lead-free solder joints," Journal of Materials Science: Materials in Electronics, vol. 30, pp. 11552-11562, 2019/06/01 2019.

4.

What do the error bars given in Figs. 11 and 12 stand for? Explain in the text.of electrical conductivity of the solder connected to cacking and/or new phases?

The error bars in the Figure 11 and Figure 12 showing the value of standard deviation. The measurements of IMC thickness at each of the samples were taken on 10 different samples at each conditions.

5.

Just a recommendation for the future. To make linear regression with only three points might be acceptable, but introducing more points is adviceble. 

Thank you for the suggestions. We will consider the suggestions for the future.

We thank you for the valuable suggestions made by the reviewers to improve our paper. All concerns and suggestions have been addressed in the above feedback. We would like to thank the editor in advance for considering our work.

Yours sincerely,

Dr Mohd Arif Anuar Mohd Salleh

Nihon Superior Electronic Material Research Lab, Center of Excellence Geopolymer & Green Technology (CeGeoGTech), Faculty of Chemical Engineering Technology, Universiti Malaysia Perlis (UniMAP), Taman Muhibbah, 02600, Arau, Perlis, Malaysia.
